# Evaluation of Antimicrobial, Antioxidant, and Cytotoxic Activity of Phenolic Preparations of Diverse Composition, Obtained from *Elaeagnus rhamnoides* (L.) A. Nelson Leaf and Twig Extracts

**DOI:** 10.3390/molecules26102835

**Published:** 2021-05-11

**Authors:** Anna Stochmal, Bartosz Skalski, Rostyslav Pietukhov, Beata Sadowska, Joanna Rywaniak, Urszula Wójcik-Bojek, Łukasz Grabarczyk, Jerzy Żuchowski, Beata Olas

**Affiliations:** 1Department of Biochemistry, Institute of Soil Science and Plant Cultivation, State Research Institute, 24-100 Pulawy, Poland; asf@iung.pulawy.pl (A.S.); jzuchowski@iung.pulawy.pl (J.Ż.); 2Department of General Biochemistry, Faculty of Biology and Environmental Protection, University of Lodz, 90-236 Lodz, Poland; bartosz.skalski@biol.uni.lodz.pl (B.S.); rrrostislav97@gmail.com (R.P.); 3Department of Immunology and Infectious Biology, Institute of Microbiology, Biotechnology and Immunology, Faculty of Biology and Environmental Protection, University of Lodz, 90-237 Lodz, Poland; beata.sadowska@biol.uni.lodz.pl (B.S.); joanna.rywaniak@biol.uni.lodz.pl (J.R.); urszula.wojcik@biol.uni.lodz.pl (U.W.-B.); 4Department of Neurology and Neurosurgery, Faculty of Medical Sciences, University of Warmia and Mazury, Warszawska 30, 10-082 Olsztyn, Poland; lukasz.grabarczyk@uwm.edu.pl

**Keywords:** antimicrobial activity, cytotoxicity, *Elaeagnus rhamnoides* (L.) A. Nelson, oxidative stress, sea buckthorn

## Abstract

Although the major components of various organs of sea buckthorn have been identified (particularly phenolic compounds), biological properties of many of these phytochemicals still remain poorly characterized. In this study, we focused on the chemical composition and biological activity of preparations that were obtained from sea buckthorn twigs and leaves. The objective was to investigate cytotoxicity of these preparations against human fibroblast line HFF-1, using MTT reduction assay, their anti- or pro-oxidant activities against the effects of a biological oxidant -H_2_O_2_/Fe—on human plasma lipids and proteins in vitro (using TBARS and carbonyl groups as the markers of oxidative stress). Antimicrobial activity of the tested preparations against Gram-positive (*Staphylococcus aureus*, *S. epidermidis*, *Enterococcus faecalis*) and Gram-negative bacteria (*Escherichia coli*, *Pseudomonas aeruginosa*), as well as against fungi (*Candida albicans*, *C. glabrata*) by the EUCAST-approved broth microdilution method, followed by growth on solid media, were also assessed. Our analysis showed significant differences in chemical composition and biological properties of the tested preparations (A–F). All tested preparations from sea buckthorn twigs (D–F) and one preparation from sea buckthorn leaves (preparation C) may be a new source of phenolic antioxidants for pharmacological and cosmetic applications.

## 1. Introduction

Sea buckthorn belongs to the *Elaeagnaceae* family. This plant has low soil requirements, dense branches, lanceolate leaves, and its characteristic feature is the presence of thorns. Moreover, the small, berry-like fruits of sea buckthorn are an intense orange color. Sea buckthorn is a rich source of many health-promoting substances, including phenolic compounds, vitamins, polysaccharides, macro- and microelements [1,2]. The antioxidant effects of sea buckthorn are derived from the presence of vitamin C, carotenoids, tocopherols, and phenolic compounds, including phenolic acids and flavonoids, which are present in different organs of this plant [1]. Our previous research showed that extracts and different fractions from this plant inhibited human plasma lipid peroxidation, plasma protein carbonylation, and protected plasma protein thiol groups from oxidation in vitro [3,4]. Hydrolysable tannins and flavonoids were main phenolics in the tested fractions from sea buckthorn leaves, while proanthocyanidins and catechin dominated in twig fraction [4].

In addition, our earlier studies indicated that this plant exerted anti-platelet activity [4]. For example, phenolic fractions from leaves and twigs, and non-polar fractions from leaves and twigs inhibited platelet aggregation and platelet adhesion to adhesive proteins, like fibrinogen and collagen [5].

Although numerous secondary metabolites of various organs of sea buckthorn have been identified (particularly phenolic compounds), biological properties of many of these phytochemicals still remain poorly characterized. In this study, we focused on chemical composition and biological activity of phenolic preparations obtained from the sea buckthorn leaf extract (fractions A, B, and C), and the sea buckthorn twig extract (fractions D, E, and F). Our aim was to investigate cytotoxicity of these preparations against the human fibroblast line HFF-1 (MTT reduction assay), and their effects on oxidative processes induced by a biological oxidant—H_2_O_2_/Fe (the donor of hydroxyl radicals)—in human plasma lipids and proteins (measured by the level of thiobarbituric acid reactive substances (TBARS), the marker of lipid peroxidation, and carbonyl groups—the marker of protein oxidation), in vitro. Antimicrobial activity of the tested preparations against selected Gram-positive bacteria (*Staphylococcus aureus*, *S. epidermidis*, *Enterococcus faecalis*), Gram-negative bacteria (*Escherichia coli*, *Pseudomonas aeruginosa*), and fungi (*Candida albicans*, *C. glabrata*) was also assessed, using the EUCAST-approved broth microdilution method, followed by growth on solid media. By conducting research on the bioactive properties of the tested preparations, it is possible to decipher the most influential classes of phenolic compounds responsible for the observed effects.

## 2. Results

### 2.1. Chemical Characterization of the Tested Preparations

The applied methods of step-gradient elution enabled us to separate the extracts of sea buckthorn leaves and twigs into fractions of very different composition. Preparations A, B, and C were obtained from the sea buckthorn leaf extract (Figure 1 and Figure 2). Preparation A was composed mainly of diverse ellagitannins; its main constituents were strictinin, casuarinin, hippophaenin, and casuarictin or their isomers. Preparation B consisted mainly of glucosides of isorhamnetin and quercetin. Isorhamnetin-3-*O*-glucoside-7-*O*-rhamnoside was the dominant compound; isorhamnetin-3-*O*-galactoside-7-*O*-rhamnoside, isorhamnetin-3-*O*-rutinoside, isorhamnetin 3-*O*-glucoside, as well as quercetin 3-*O*-glucoside, quercetin 3-*O*-hexoside-deoxyhexosides (probably of quercetin 3-*O*-Hex-7-*O*-dHex structure), and rutin were other major flavonoids. In addition, preparation B contained ellagic acid, ellagic acid hexoside, and ellagic acid pentoside. In contrast, preparation C consisted mainly of acylated glycosides of isorhamnetin, kaempferol, and quercetin, such as kaempferol 3-*O*-hexosides acylated with *p*-coumaric acid, and isorhamnetin or quercetin 3-hexosides-deohyhexosides acylated with (-)-linalool-1-oic acid (a monoterpenoid), or its isomer. 

The remaining 3 fractions were obtained from sea buckthorn twig extract (Figure 1 and Figure 2). Preparation D consisted of (epi)gallocatechin, and (epi)gallocatechin-containing dimeric and trimeric B-type proanthocyanidins. Catechin, and diverse dimeric, trimeric, and tetrameric B-type proanthocyanidins, composed of (epi)catechin units were major constituents of preparation E. It contained smaller amounts of casuarinin/isomer, ellagic acid hexoside, and ellagic acid pentoside. Preparation F contained ellagic acid, ellagic acid pentoside, and different flavonoids (isorhamnetin-3-*O*-glucoside-7-*O*-rhamnoside, quercetin-3-*O*-glucoside, isorhamnetin-3-*O*-rutinoside, isorhamnetin-3-*O*-glucoside, kaempferol-coumaroyl hexoside, free flavonol aglycones). Other phenolic compounds, showing UV maxima at about 260–300 nm, were also present. Non-phenolic constituents were also detected, such as a putative triterpenoid hexoside (C_30_H_48_O_6_-hexose; *m/z* 711−a formic acid adduct) and an unidentified compound showing a deprotonated ion at *m/z* 327. 

### 2.2. Cytotoxicity of the Preparations from Sea Buckthorn Twigs and Leaves

Cytotoxic effect of the tested sea buckthorn preparations was assessed against human fibroblasts line HFF-1. The fibroblasts are recommended as one of the lines for in vitro cytotoxicity testing in ISO 10993-5:2009 [6]. Usually, mouse fibroblasts line L929 are used, however, as we assume the use of sea buckthorn preparations as potential biologically active substances in pharmacologic or cosmetic products, we decided to use human fibroblasts line HFF-1. As presented in Figure 3, preparations B and F did not affect the fibroblasts growth at the whole concentration range tested. The viability of the cells was from 90.3 ± 9.9% to 106.3 ± 2.1% and from 87.3 ± 13.9% to 104.3 ± 0.6% after 24 h exposition on preparation B and F, respectively. Preparations C and E were cytotoxic for human fibroblasts line HFF-1 at the highest concentration used (1000 µg/mL), decreasing cell viability to 64.1 ± 7.5% (C) and to 23.4 ± 5.2% (E). The strongest cytotoxic effect was expressed by preparations A and D when used at a concentration above 500 µg/mL. The viability of the fibroblasts did not exceed 29% and 12% in the presence of preparations A and D used at two of the highest concentrations tested, respectively. However, both preparations used at a concentration range of 3.9–250 µg/mL did not adversely affect the cells’ viability.

### 2.3. Effects of Phenolic Preparations on Oxidative Stress Biomarkers in Human Plasma In Vitro

Pro-oxidant or antioxidant properties of six preparations from *E. rhamnoides* (L.) A. Nelson leaves and twigs (at a dose range 5–50 µg/mL) were studied in an in vitro model using human plasma, which was exposed to H_2_O_2_/Fe (the donor of OH^∙^). None of the preparations altered the level of biomarkers of oxidative stress in plasma not treated with H_2_O_2_/Fe (data not shown). As demonstrated in Figure 4 and Figure 5, exposure of plasma to H_2_O_2_/Fe resulted in an enhanced level of various biomarkers of oxidative stress, including the level of TBARS—marker of lipid peroxidation, and protein carbonylation—marker of protein damages (Figure 4 and Figure 5). At the highest concentration—50 µg/mL), all used preparations from leaves and twigs, including A, caused statistically significant reduction of lipid peroxidation induced by H_2_O_2_/Fe, as compared to the control sample (plasma treated only with H_2_O_2_/Fe). Preparations B–F inhibited this process at the concentration of 10 µg/mL (Figure 4). Moreover, preparation C from sea buckthorn leaves and all preparations from sea buckthorn twigs (D–F) were shown to protect plasma proteins against H_2_O_2_/Fe-induced protein carbonylation, at concentrations 10 and 50 µg/mL (Figure 5).

Table 1 demonstrates comparative effects of the six preparations (A–F) from sea buckthorn leaves and twigs concentration at –10 µg/mL) on two selected parameters of oxidative stress: lipid peroxidation and protein carbonylation, in plasma treated with H_2_O_2_/Fe. Four tested preparations (C–F) had stronger antioxidant activity than two other used preparations (A and B). For example, preparations C–F inhibited lipid peroxidation and protein carbonylation (Table 1). 

### 2.4. Antimicrobial Activity of Phenolic Preparations from Sea Buckthorn Twigs and Leaves

Antimicrobial effect of fractionated sea buckthorn preparations was presented in Table 2 as minimum inhibitory and bactericidal/fungicidal concentrations (MIC and MBC/MFC) against a broad panel Gram-positive and Gram-negative reference bacterial strains, as well as against fungi from *Candida* sp.

Preparations A, D, and E seemed to be the most potent antimicrobials, exhibiting mainly anti-staphylococcal activity at quite low concentrations (mostly 125–500 µg/mL). Sensitivity of staphylococci on these preparations also depended on the strain used. The growth of *S. epidermidis* was the strongest inhibited, however, biocidal effect was shown only at 8–16-fold higher concentration of these preparations. It is worth noticing that preparations A and D used at a concentration of 500 µg/mL were also active against *P. aeruginosa*, which belongs to bacteria widespread in nature and, thus, resistant to environmental stress conditions. The preparations C and F were inactive against bacteria in the whole range of concentrations tested, except for preparation F against *S. epidermidis* with MIC at 1000 µg/mL. The tested preparations did not affect the growth and viability of *Candida* yeasts up to 1000 µg/mL.

## 3. Discussion

Our experiments showed that the tested preparations of different chemical compositions also differed significantly in their biological properties. Antimicrobial activity of our novel polyphenolic preparations made in a sequential manner from the full extract was similar to antimicrobial effects exerted by polyphenolic leaf and twig extracts in our previous study [7]. Preparations A–F were inactive against Gram-negative bacteria, such as *E. coli* and *P. aeruginosa* (with an exception of preparations A and D), as well as against fungi from *Candida* sp. (Table 2) as previously tested phenolic and non-polar fractions of *E. rhamnoides* leaf and twig extracts. However, leaf preparation A showed four times higher activity against *S. aureus* ATCC 43300 in comparison to previously tested phenolic fractions of leaf extract (LF) (MIC A = 125 µg/mL vs. MIC LF = 500 µg/mL). Similarly, twig preparations D and E exhibited four times and two times, respectively, stronger effects against *S. aureus* ATCC 29213 than a previously tested phenolic fraction of twig extract (GF) (MIC D = 250 µg/mL and MIC E = 500 µg/mL vs. MIC GF = 1000 µg/mL) [7]. In the literature, the relationship between the chemical compositions of plant origin extracts/fractions/preparations and their biological activities was generally speculated [8,9]. Nevertheless, the changes observed here are not so easy to explain, since, for instance, preparation A was rich in ellagitannins (Figure 1), while hydrolysable tannins and ellagic acid were also dominant compounds (above 31%) of previously tested LF [7]. Likewise twig preparations D and E, mainly containing proanthocyanidins, such as catechin and epigallocatechin (Figure 2), seem to be similar in chemical composition to previously tested GF with 47.5% of proanthocyanidins and catechin [7]. Moreover, the preparations were not equally active against different microbial strains from even the same species (e.g., MIC A against *S. aureus* ATCC 29213 = 1000 µg/mL) suggesting strain-dependent susceptibility on such preparations. Such results pose difficulties in registering of plant preparations as antimicrobials for medical use. On the other hand, we demonstrated that the lowest concentration of tested *E. rhamnoides* preparations inhibiting microbial growth (62 µg/mL—MIC A against *S. epidermidis*) was still higher than concentrations at which these preparations reduced H_2_O_2_/Fe-induced lipid peroxidation and protein carbonylation (10 and 50 µg/mL, Figure 4 and Figure 5). Thus, the tested preparations could be safely administered orally as antioxidants without affecting gut microbiota. 

In our present experiments, human plasma was incubated with phenolic preparations (A–F) in three concentrations (5, 10, and 50 µg/mL) and two lower concentrations (5 and 10 µg/mL) appear to correspond to the physiological concentrations available after oral administrated [10,11]. Our present findings confirm that the six tested preparations (A–F, especially at the highest used concentration of 50 µg/mL) isolated from sea buckthorn leaves and twigs demonstrated antioxidative potential in an in vitro human plasma treated with H_2_O_2_/Fe: tested preparations (A–F) inhibited plasma lipid peroxidation induced by H_2_O_2_/Fe, as measured by the level of TBARS. Moreover, the tested preparations (A–F) reduced plasma protein carbonylation induced by H_2_O_2_/Fe. However, they had different influences on the oxidative stress in human plasma treated with H_2_O_2_/Fe, which may be attributed to the differences in their chemical content. Four preparations seemed to offer the most promise (one preparation from leaves (preparation C) and three preparations from twigs (D–F)). Preparation C consisted mainly of acylated glycosides of isorhamnetin, kaempferol, and quercetin; preparation D consisted of (epi)gallocatechin, and (epi)gallocatechin-containing dimeric and trimeric B-type proanthocyanidins; preparation E consisted of catechin, and diverse dimeric, trimeric, and tetrameric B-type proanthocyanidins; preparation F contained ellagic acid, ellagic acid pentoside, and different flavonoids. Similar effects were observed in other experiments. For example, isorhamnetin and its derivatives isolated from sea buckthorn leaves had antioxidant potential [2]. Moreover, results by Sun et al. [12] indicate that isorhamnetin inhibits H_2_O_2_ action by scavenging reactive oxygen species. However, used preparation D isolated from twigs, in particular, had stronger antioxidant properties than preparations E and F. For example, inhibition of lipid peroxidation was about 45% for preparation D (10 µg/mL), and about 35% for preparation E (10 µg/mL). Strong antioxidant properties of preparation D may depend on the presence of proanthocyanidins, which was demonstrated to be one of the most powerful natural antioxidants. It is worth pointing out that all tested preparations had no cytotoxic effect against human fibroblasts, up to a concentration of 250 µg/mL, which authorizes their use in vivo at the above-described active concentrations.

In conclusion, all tested preparations from sea buckthorn twigs (D–F) and one preparation from sea buckthorn leaves (preparation C) may be new sources of phenolic antioxidants for pharmacological and cosmetic applications.

## 4. Material and Methods

### 4.1. Chemicals

Dimethyl sulfoxide (DMSO), thiobarbituric acid (TBA), formic acid (LC-MS grade), and H_2_O_2_ were purchased from Sigma-Merck (St. Louis, MO, USA). Methanol (isocratic grade) and acetonitrile (LC-MS grade) were acquired from Merck (Darmstadt, Germany). Other reagents represented analytical grades and were provided by commercial suppliers, including POCh, (Poland), Acros (Poland), and Chempur (Poland). 

### 4.2. Plant Material

Sea buckthorn twigs and leaves were obtained from a horticultural farm in Sokółka, Podlaskie Voivodeship, Poland (53°24′ N, 23°30′ E)—the greatest Polish producer of sea buckthorn fruits. The plant material was identified by Mr. Stanislaw Trzonkowski, the owner of the farm. Voucher specimens were deported at the Institute of Soil Science and Plant Cultivation−Sate Research Institute, Pulawy, Poland (IUNG/HRH/2015/2).

### 4.3. Preparation of Phenolic Preparations from Sea Buckthorn Leaves and Twigs

Butanol extracts from leaves and twigs of sea buckthorn were prepared according to previously described methods [7,13]. Briefly, freeze-dried sea buckthorn leaves were powdered in a laboratory mill (ZM200, Retsch, Haan, Germany). A 140 g portion of the powdered leaves was extracted with 3L (in three portions) of 80% methanol (*v/v*), at room temperature, for 48 h; the extraction was supported by ultrasonication (6 × 10 min). The extracts were filtered, concentrated in a rotary evaporator, and extracted with hexane. The defatted extract was evaporated in a rotary evaporator to remove organic solvents; the residue was resuspended in Milli-Q water, acidified with formic acid, and subjected to *n*-butanol extraction. The butanol extract was evaporated in a rotary evaporator, and the residue was suspended in Milli-Q water (20% *t*-butanol was also used to dissolve from the evaporation flask), and freeze-dried [13]. The twigs were air dried in a laboratory drier (40 °C), and ground in laboratory mills (SM300, ZM200, Retsch). The ground twigs (680 g) were extracted with 14 L (in three portions) of 80% methanol (*v/v*), at room temperature, for 48 h; the extraction was supported by ultrasonication (3 × 10 min). The remaining part of the procedure was the same as the one described above [7].

A portion of the butanol extract of sea buckthorn leaves (3.14 g) was dissolved in 200 mL of 1.5% methanol + 0.1% formic acid, and sonicated for 5 min. The mixture was centrifuged, and the supernatant was loaded onto a C18 column (65 × 70 mm; COSMOSIL 140C18-Prep, 140 µm), equilibrated with 1.5% methanol + 0.1% formic acid. The column was washed with 400 mL of the same solvent, to remove the most polar compounds. The compounds bound to the column were subsequently eluted with methanol solutions of increasing concentration: 10%, 30%, 66% and 80%. The eluates were evaporated in a rotary evaporator and lyophilized, to yield 0.076 g of 10% methanol fraction, 1.427 g of 30% methanol fraction (preparation A), 0.505 g of 50% methanol fraction (preparation B), 0.241 g of 66% methanol fraction (preparation C), and 0.118 g of 80% methanol fraction. The twig extract was fractionated in a similar way. A 3.68 g portion was dissolved as described above, and loaded onto a C18 column (34 × 100 mm; COSMOSIL 140C18-Prep, 140 µm), equilibrated with 1.5% methanol + 0.1% formic acid. The column was washed with 400 mL of the same solvent, and bound compounds were eluted using a step gradient of increasing methanol concentrations: 10% (500 mL), 40% (500 mL), 70% (50 mL) and 85% (500 mL). The eluates were evaporated in a rotary evaporator and lyophilized, to yield 0.515 g of 10% methanol preparation (preparation D), 2.067 g of 40% methanol preparation (preparation E), 0.135 g of 70% methanol preparation (preparation F), and 0.003 g of 85% methanol preparation. A diagram of the applied methods is shown in Figure 6.

### 4.4. LC-MS Analysis

UHPLC-MS analyses of the investigated preparations (A–F) were carried out using an ACQUITY UPLC™ system (Waters, Milford, MA, USA), hyphenated with an ACQUITY TQD (Waters) mass spectrometer, according to the previously described procedure [14]. An ACQUITY BEH C18 (100 mm × 2.1 mm, 1.7 μm; Waters) column was applied. The column was maintained at 50 °C, the flow rate was 0.500 mL min^−1^, and the injection volume was 2.5 µL. The mobile phase consisted of mixtures of solvent B (acetonitrile with 0.1% of formic acid) in solvent A (0.1% solution of formic acid in Milli-Q water), the following elution method was applied: 0–0.5 min: 7% B; 0.5–11.90 min: 7–80% B (a linear gradient); 12–13 min: 99% B; 13.10–15 min: 7% B. The TQ mass spectrometer was operated in negative and positive ion mode. Details of MS setting can be found in the work of Żuchowski et al. [14]. Constituents of the analyzed fractions were identified on the basis of their MS and UV spectra, with the help of data from our previously performed LC–HRMS/MS (Q-TOF) analyses of phenolic fractions of sea buckthorn leaves and twigs [4], including unpublished data, as well as available literature [13,14,15,16].

### 4.5. Stock Solutions

Stock solutions of preparations (A–F) from sea buckthorn twig and leaves were made in 50% DMSO (for measuring oxidative stress in human plasma). The final concentration of DMSO in samples was lower than 0.05% and its effects were determined in all experiments.

### 4.6. Bacterial Strains and Culture

Reference bacterial strains: *Staphylococcus aureus* ATCC 29213 (MSSA, methicillin-sensitive strain), *S. aureus* ATCC 43300 (MRSA, methicillin-resistant strain), *S. epidermidis* ATCC 12228, *Enterococcus faecalis* ATCC 29212, *Escherichia coli* ATCC 25922, *Pseudomonas aeruginosa* NCTC 6749, and fungi: *Candida albicans* ATCC 10231, *C. glabrata* ATCC 90030 were used in this study. Stock cultures were kept frozen at −80 °C in tryptic soy broth (TSB) with 15% glycerol (bacteria) or in peptone yeast extract glucose broth (PYG) with 15% glycerol (fungi). The ready-to-use cultures were freshly prepared on tryptic soy agar (TSA) or Sabouraud agar (SDA), for bacteria or fungi, respectively. All culture media were purchased from BTL (Poland). 

### 4.7. Cytotoxicity of Sea Buckthorn Preparations

Human foreskin fibroblasts (HFF-1) were cultured in Dulbecco’s Modified Eagle’s Medium high glucose (DMEM hg; Biowest, MO, USA), supplemented with 15% fetal bovine serum (FBS; Biological Industries, Cromwell, CT, USA) and penicillin/streptomycin (100-fold concentrate; Biological Industries, Cromwell, CT, USA). Briefly, a detached cell suspension (2 × 10^5^ cells/mL) was seeded at 100 µL/well into 96-well tissue culture plates (Nunc, Denmark) for 24 h at 37 °C, 5% CO_2_. The culture medium was replaced with 100 µL medium containing fractionated sea buckthorn preparations at a range of 3.9–1000 µg/mL for 24 h. The preparations were initially dissolved in sterile water for injection (Sigma-Merck, St. Louis, MO, USA) (A,D) or 100% methanol (Me-OH; POCH, Gliwice, Poland) (B,C,E,F) to obtain stock solutions 4 mg/ml and 40 mg/mL, respectively. Then, the tested preparations were diluted in cell culture medium to the concentration range tested. The final highest concentration of Me-OH in samples B–C and E–F was 2.5%; therefore, appropriate cell growth control (HFF-1 in culture media alone and in culture media with 2.5% Me-OH) were included. The cytotoxicity was measured by MTT-reduction assay (50 µL MTT at 1.5 mg/mL per well was used) according to Müller and Kramer [17] and ISO 10993-5:2009 [6] in its own modification. The absorbance (A_550_) of the samples was assessed using a microplate reader (Victor2, Wallac, Finland), and the percentage of viable cells in comparison to the growth control was calculated.

### 4.8. Human Plasma Isolation

Fresh human plasma was obtained from regular, medication-free donors of a blood bank at a Medical Center (Lodz, Poland). Peripheral blood was also obtained from non-smoking men and women (collected into CPD solution (citrate/phosphate/dextrose; 9:1; *v/v* blood/CPD) or CPDA solution (citrate/phosphate/dextrose/adenine; 8.5:1; *v/v*; blood/CPDA)). They had not taken any medication or addictive substances (including tobacco, alcohol, antioxidant, or supplementation). Our analysis of the blood samples was performed under the guidelines of the Helsinki Declaration for Human Research, and approved by the Committee on the Ethics of Research in Human Experimentation at the University of Lodz (resolution No. 3/KBBN-UŁ/II/2016). Plasma was pre-incubated (5 min, at 37 °C) with preparations from the leaves (A,B,C) and twigs (D,E,F) of sea buckthorn at the final concentrations of 5–50 μg/mL and then 4.7 mM H_2_O_2_/3.8 mM Fe_2_SO_4_/2.5 mM EDTA (25 min, at 37 °C).

Plasma protein concentration, determined by measuring absorbance at 280 nm (in the tested samples), was calculated according to the procedure of Whitaker and Granum [18].

### 4.9. Markers of Oxidative Stress

#### 4.9.1. Plasma Lipid Peroxidation

Lipid peroxidation level was determined by measuring of the concentration of TBARS. Incubation of plasma (control, plant preparation and H_2_O_2_/Fe-treated plasma) was stopped by cooling the samples in an ice bath. Samples of plasma were transferred to an equal volume of cold 15% (*v/v*) trichloroacetic acid in 0.25 M HCl and 0.37 M thiobarbituric acid in 0.25 M HCl, immersed in a boiling water bath for 10 min, and then centrifuged at 10,000× *g* for 15 min, 18 °C. Absorbance was measured at 535 nm (the SPECTROstar Nano Microplate Reader from BMG LABTECH Germany) [8,19]. The TBARS concentration was calculated using the molar extinction coefficient (ε = 156,000 M^−1^ cm^−1^).

#### 4.9.2. Carbonyl Group Measurement

The detection of carbonyl groups in plasma proteins was carried out according to Levine et al. and Bartosz [8,9]. The carbonyl group concentration was calculated using a molar extinction coefficient (ε = 22,000 M^−1^ cm^−1^), and the level of carbonyl groups was expressed as nmol carbonyl groups/mg of plasma protein. Carbonyl content was determined with the use of SPECTROstar Nano Microplate Reader from BMG LABTECH Germany. 

### 4.10. Antimicrobial Activity

Minimum inhibitory concentration (MIC) of *E. rhamnoides* fractionated leaf and twig preparations against bacteria (*n* = 6) and fungi (*n* = 2) was assessed twice using the broth microdilution assay recommended by EUCAST [20]. The preparations of sea buckthorn were initially dissolved, as described in Section 4.7. Then, the tested preparations were diluted in Mueller–Hinton Broth (Grasso, Poland) for bacteria or RPMI-1640 (Sigma-Merck, St. Louis, MO., USA) with 2% glucose for fungi to the final concentration range of 1000–15.6 µg/mL. Bacterial suspensions prepared at a density 2–8 × 10^5^ CFU/mL or fungal suspensions at a density of 5 × 10^5^ CFU/ml were added (*v/v*, 1:1) to the dilution series of the preparations. The final highest concentration of Me-OH in samples B–C and E–F was 2.5%; therefore, appropriate positive growth control (microbial suspensions in culture media alone and in culture media with 2.5% Me-OH) were included. The MIC of the extracts was defined after 24 h incubation at 37 °C as the lowest concentration causing the inhibition of microbial growth in comparison to a positive control. To determine minimum bactericidal/fungicidal concentrations (MBC/MFC), 10 µL of the samples indicated as MIC, and these, with two higher concentrations, were cultured for 24 h at 37 °C on TSA or SDA, for bacteria and fungi, respectively. The concentration causing 99.9% of microbial killing was defined as MBC/MFC. The test was performed twice in two replicates. 

### 4.11. Data Analysis

In order to eliminate uncertain data, the Q-Dixon test was performed. All of the values in this study were expressed as mean ± SD. The obtained results were firstly analyzed under the account of normality with the Shapiro–Wilk test and equality of variance with the Levine test. Statistically significant differences were assessed by applying the ANOVA test (the significance level was *p* < 0.05), followed by Tukey’s multiple comparison test or the Kruskal–Wallis test.

## Figures and Tables

**Figure 1 molecules-26-02835-f001:**
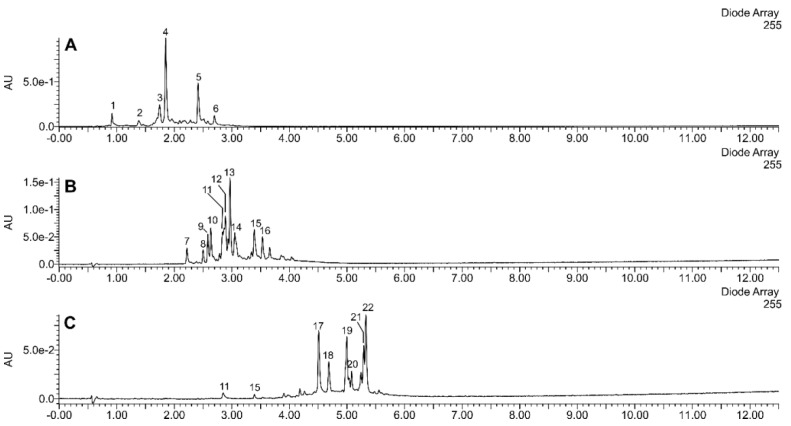
UHPLC-UV chromatogram of preparations (**A**–**C**), made from the extract of sea buckthorn leaves. Major peaks: 1—pedunculagin/isomer; 2—pedunculagin/isomer; 3—isostrictinin/isomer; 4—casuarinin and hippophaenin/isomers; 5—casuarictin/isomer; 6—ellagitannin C_46_H_30_O_30_; 7—ellagic acid-Hex; 8—Q-Hex-dHex; 9—Q-Hex-dHex; 10—ellagic acid-Pen; 11—ellagic acid; 12—I-3-*O*-Gal-7-*O*-Rha; 13—I-3-*O*-Glc-7-*O*-Rha; 14—Q-3-*O*-Glc; 15—I-3-*O*-Rut; 16—I-3-*O*-Glc; 17 and 18—K-Hex-CouA; 19—Q-Hex-dHex-Lin; 20—Q-Hex-Hex-dHex-LinA-FerA; 21 and 22—I-Hex-dHex-LinA. I−isorhamnetin; K−kaempferol; Q−quercetin; Gal−galactose; Glc−glucose; Rha−rhamnose; Rut–rutinose; Hex−hexose; dHex−deoxyhexose; Pen−pentose; CouA−coumaric acid; FerA−ferulic acid; LinA−(-)-linalool-1-oic acid/isomer (C_10_H_16_O_3_).

**Figure 2 molecules-26-02835-f002:**
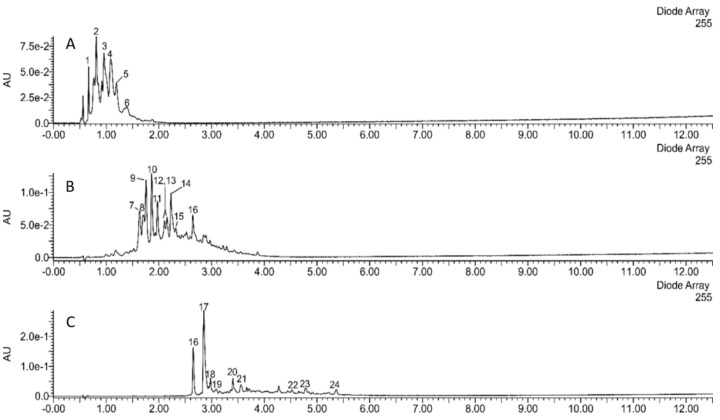
UHPLC-UV chromatogram of preparations (**A**–**C**), made from the extract of sea buckthorn twigs. Major peaks: 1—GalA-Hex; 2—(epi)Gc-(epi)Gc; 3—(epi)Gc, (epi)Gc-(epi)Gc; 4—(epi)C-(epi)Gc; 5—(epi)C-(epi)Gc; 6—(epi)C-(epi)C-(epi)Gc; 7—(epi)C-(epi)C; 8—(epi)C-(epi)C-(epi)C; 9—catechin; 10—casuarinin/isomer; 11—(epi)C-(epi)C-(epi)C-(epi)C; 12 and 13—(epi)C-(epi)C-(epi)C; 14—(epi)C-(epi)C; 15—(epi)C-(epi)C-(epi)C-(epi)C; 16—(epi)C-(epi)C-(epi)C; ellagic acid-Pen; 17—ellagic acid; 18—I-3-*O*-Glc-7-*O*-Rha; 19—Q-3-*O*-Glc; 20—I-3-*O*-Rut; 21—I-3-*O*-Glc; 22—K-Hex-CouA; 23—I-dHex; 24—I. (epi)Gc—(epi)gallocatechin; (epi)C—(epi)catechin; I—isorhamnetin; K—kaempferol; Q—quercetin; Glc—glucose; Rha—rhamnose; Rut—rutinose; Hex—hexose; dHex—deoxyhexose; Pen—pentose; CouA—coumaric acid; GalA—gallic acid.

**Figure 3 molecules-26-02835-f003:**
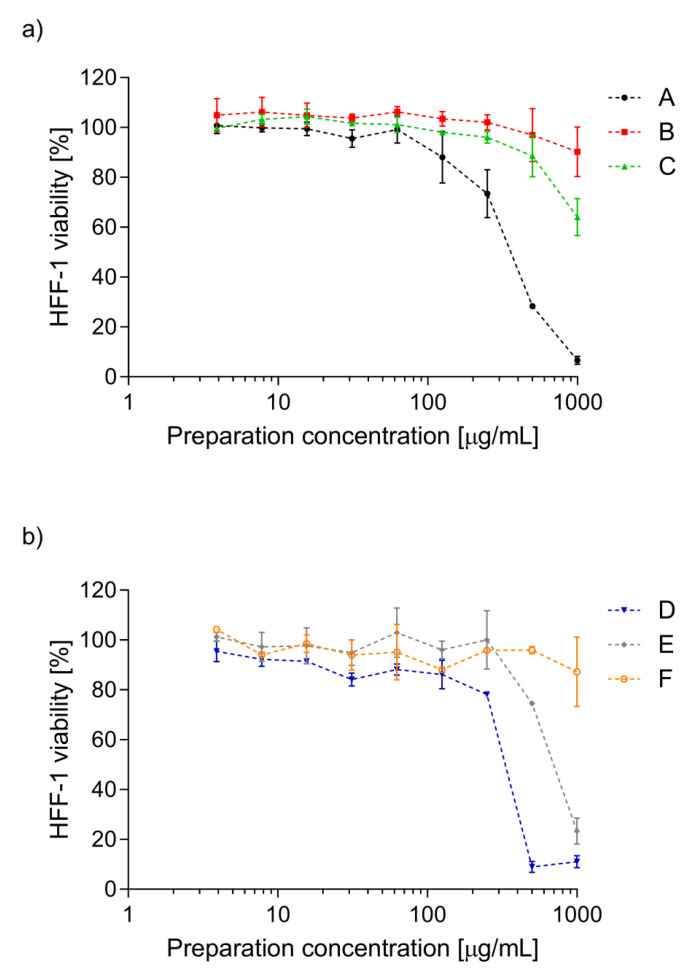
The viability of human foreskin fibroblasts line HFF-1 exposed (24 h) on the preparations (A–F) of *E. rhamnoides* (L.) A Nelson leaves (**a**) and twigs (**b**) (3.9–1000 µg/mL). Data represent means ± SD of two independent experiments in duplicates.

**Figure 4 molecules-26-02835-f004:**
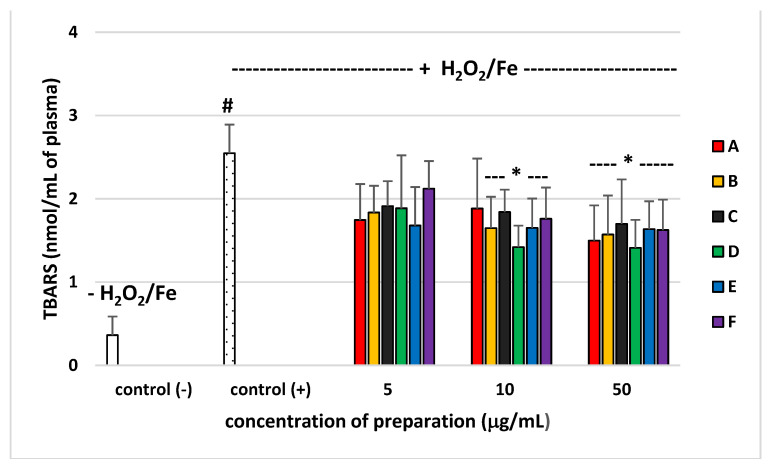
Effects of the preparations (A–F) of *E. rhamnoides* (L.) A. Nelson leaves and twigs (5–50 µg/mL; 30 min) on plasma lipid peroxidation induced by H_2_O_2_/Fe. Data represent means ± SD of six independent experiments. * *p* < 0.05 (vs. control (+)), # *p* < 0.01 (vs. control (−)). Control negative (−) refers to plasma not treated with H_2_O_2_/Fe, whereas control positive (+) refers to plasma treated with H_2_O_2_/Fe.

**Figure 5 molecules-26-02835-f005:**
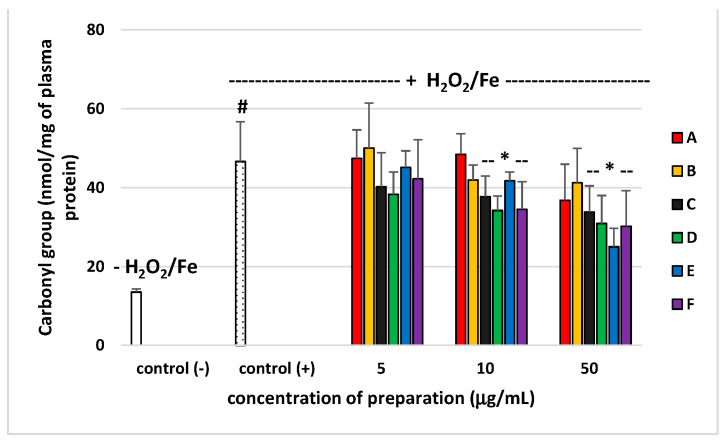
Effects of the preparations (A–F) of *E. rhamnoides* (L.) A. Nelson leaves and twigs (5–50 µg/mL; 30 min) on plasma protein carbonylation induced by H_2_O_2_/Fe. Data represent means ± SD of six independent experiments. * *p* < 0.05 (vs. control (+)), # *p* < 0.01 (vs. control (−)). Control negative (−) refers to plasma not treated with H_2_O_2_/Fe, whereas control positive (+) refers to plasma treated with H_2_O_2_/Fe.

**Figure 6 molecules-26-02835-f006:**
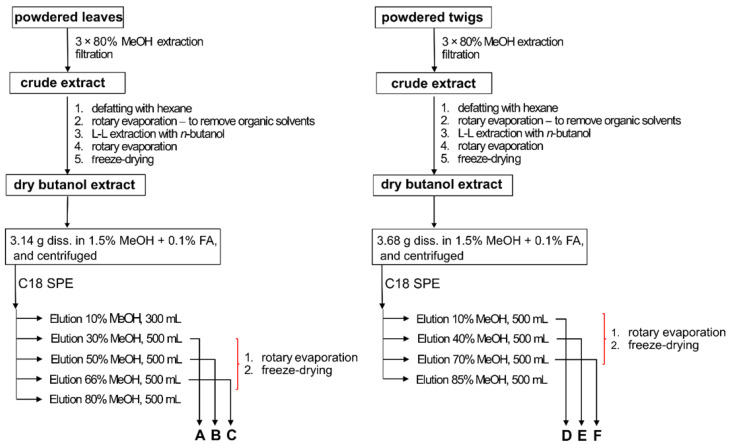
A diagram of methods of extraction and fractionation applied to obtain the investigated phenolic preparations of sea buckthorn leaf extract (**A**–**C**) and twig extract (**D**–**F**).

**Table 1 molecules-26-02835-t001:** Antioxidant activity of phenolic preparations (A–F; 10 µg/mL) from *E. rhamnoides* leaves (A–C) and twigs (D–F).

Oxidative Stress	Preparation					
	A	B	C	D	E	F
**Lipid peroxidation**	No effect	inhibition	inhibition	inhibition	inhibition	inhibition
**Protein carbonylation**	No effect	No effect	inhibition	inhibition	inhibition	inhibition

**Table 2 molecules-26-02835-t002:** Antimicrobial activity of phenolic preparations (A–F) from *E. rhamnoides* leaves (A–C) and twigs (D–F).

Microorganism	MIC [µg/mL] MBC/MFC [µg/mL]					
	A	B	C	D	E	F
**Bacteria**						
*Staphylococcus aureus* ATCC 29213	1000	1000	>1000	250	500	>1000
	1000	>1000	>1000	250	500	>1000
*Staphylococcus aureus* ATCC 43300	125	1000	>1000	500	250	>1000
	250	1000	>1000	1000	1000	>1000
*Staphylococcus epidermidis* ATCC 12228	62	500	>1000	125	125	1000
	1000	>1000	>1000	1000	1000	>1000
*Enterococcus faecalis* ATCC 29212	1000	>1000	>1000	1000	1000	>1000
	>1000	>1000	>1000	>1000	>1000	>1000
*Escherichia coli* ATCC 25922	>1000	>1000	>1000	1000	>1000	>1000
	>1000	>1000	>1000	>1000	>1000	>1000
*Pseudomonas aeruginosa* NCTC 6749	500	>1000	>1000	500	>1000	>1000
	500	>1000	>1000	500	>1000	>1000
**Fungi**						
*Candida albicans* ATCC 10231	>1000	>1000	1000	>1000	>1000	>1000
	>1000	>1000	>1000	>1000	>1000	>1000
*Candida glabrata* ATCC 90030	>1000	>1000	>1000	>1000	>1000	>1000
	>1000	>1000	>1000	>1000	>1000	>1000

Minimum inhibitory and bactericidal/fungicidal concentration (MIC and MBC/MFC) were measured by broth microdilution assay followed by the culture on solid media.

## Data Availability

All data are available from the authors.

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
