# Peer review of "Evaluation of Antimicrobial, Antioxidant, and Cytotoxic Activity of Phenolic Preparations of Diverse Composition, Obtained from Elaeagnus rhamnoides (L.) A. Nelson Leaf and Twig Extracts"

_molecules, 2021, doi:10.3390/molecules26102835_

Round 1
Reviewer 1 Report
General Comment
The manuscript of Anna Stochmal, Bartosz Skalski, Rostyslav Pietukhov, Beata Sadowska, Joanna Rywaniak, Urszula Wójcik-Bojek, Lukasz Grabarczyk, Jerzy Żuchowski, Beata Olas *entitled: “A comparison of the biological properties of phenolic preparations with different chemical content from Elaeagnus rhamnoides (L.) A. Nelson leaves and twigs“ submitted to Molecules journal, presents their work in the field of natural products chemistry. Precisely, human blood plasma was used as in vitro model to study potential protective effects of differently produced polyphenolic preparations from leafs and twigs against lipid peroxidation and protein carbonylation events caused by excessive ROS (OH radical) production with H2O2/Fe2+ Fenton-like system. In addition, authors assessed survival of human foreskin fibroblasts in cell culture via MTT assay and microbiological resistance on several ATCC strains of pathogenic and normal skin flora bacteria and fungi. The sole novelty is related to the production of 3 different polyphenolic preparations from leaves and twigs each (6 extracts in total). The other methods and approaches were elaborated in their previous research, and were used here, which is important knowledge addition, if the aim was to decipher which classes of phenolic compounds are more responsible for certain bio-actions or effects under consideration.
However, there are some necessary improvements, related to brevity and clarity of manuscript, methodological issues and flow & quality of discussion.
Major Comments
- While the overall research design is appropriate, there is a room for improvement in terms of clarity, scientific methodology, data/results presentation and contribution to discussion via focused and brave synthetic approach. The best way to address this comment is to respond to all specific and minor comments outlined below.
- Style of the English language. The same effort that authors exerted by doing the experimental part, should be repeated when writing the manuscript. Frequently I face decent experimental works and setups, and poor writing. Inadequate writing negatively influences understanding (sense that was aimed to be conveyed to reading public), and meaning as well. For example, when expressing an idea in a sentence and then backing up with references, this expression should be specific as possible, informative and still brief. If possible, engage English language editor with life science background to correct grammar and style based on these recommendations and major study aim. Throughout the manuscript there are needed repairs, I give one example here, as the second sentence from discussion: “Considering direct antimicrobial activity of tested preparations from sea buckthorn leaves and twigs in the context on our previous study on biological activity of phenolic and non-polar fractions obtained from the same plant material [7] no spectacular differences were found.” It seems like you disrespect your own work. Instead it can be expressed like this:” Antimicrobial activity of our novel polyphenolic preparations made in sequential manner from the full extract was similar to antimicrobial effects exerted by polyphenolic leaf and twig extracts in our previous study (alternatively here could be mentioned the difference in preparation from the current study and the previous one, so that authors and reading public has a chance to conclude something sensibly).
- Abstract should not be written as laboratory diary. A reader does not care about polyphenolic preparations labelled as A. B C d E F, but care to know how they are different and from what plant source they stem.
- Introduction “Although the major components (especially phenolic compounds) of various organs of sea buckthorn have been described, biological properties of many of these phytochem-icals still remains unexplored. In this study, we focused on chemical composition and bi-ological activity of three phenolic preparations (A, B, and C), which were isolated from the extract of sea buckthorn leaves, and three phenolic preparations (D, E, and F), which were obtained from sea buckthorn twig extract. The objective was to investigate cytotox-icity of these preparations against human fibroblasts line HFF-1 using MTT reduction as-say, their anti- or prooxidant activity against the effect of a biological oxidant - H2O2/Fe (the donor of hydroxyl radicals) on human plasma lipids and proteins (measured by the level of thiobarbituric acid reactive substances (TBARS, the marker of lipid peroxidation and carbonyl groups – the marker of protein oxidation) in vitro. Antimicrobial activity of tested preparations against Gram-positive (Staphylococcus aureus, epidermidis, Enterococ-cus faecalis) and Gram-negative bacteria (Escherichia coli, Pseudomonas aeruginosa), as well as against fungi (Candida albicans, C. glabrata) by EUCAST approved broth microdilution method followed by growth on solid media was also assessed.”
It is not clear what was the reason that drove authors to conduct new study on the same parts and substrate, e.g. plant sea buckthorn leaves and twigs, with the same methodology. What was the rationale for the objective stated? What was the hypothesis? This should be rewritten. I already gave an idea how to do it, please see end of comment no 2. The rationale/hypothesis could be:
If very well characterized sub-fractions of phenolic extracts are made with contents conferring high specifities/uniqueness, than by assessing simultaneously their bio active properties such as pro or antioxidant and antimicrobial effects, it is possible to decipher the most impactful classes of phenolic compounds responsible for the effects observed.
- Methodology 1– The most important concern here is weather authors used simultaneous incubation of human blood plasma with artificial Fenton-like system with H2O2/Fe2+ and polyphenolic extracts. If this is the situation, then they should read this publication “Enhanced Degradation of Phenol by a Fenton-LikeSystem (Fe/EDTA/H2O2) at Circumneutral pH” from MDPI Catalysts journal https://www.mdpi.com/2073-4344/9/5/474 and amend manuscript accordingly either with proper experimental control or by elaborating it in discussion based on their previous or others results. It might be that their results could be even more pronounced if there was no decomposition of certain polyphenolics caused by acidic to neutral pH in combination with 4.7 mM H2O2/3.8 mM Fe2SO4/2.5 mM EDTA.
- Methodology 2 – if authors stated that they aim to examine prooxidant effects of phenolics from leaves and twigs, why they haven’t included treatment comprised of solely plasma and phenolic extract incubated for 30 min on 37 degree and run TBARS test to look for certain prooxidant effects. There are of course other experimental options to do so. Please check paper https://www.hindawi.com/journals/jt/2011/467305/ for this venue. To be clear enough, the proper group setup beside those shown in figure 4 and 5 should be 1) only plasma, 2) plasma and phenolic extracts, 3) plasma with Fenton like system, 4) plasma with Fenton like system and extracts. Perhaps authors checked this aspect previously; if so certain amendments and elaborations should be included in the new manuscript version. Finally, authors could substantially improve their study by assessing individual human plasma with the most effective phenolic preparations, to contribute with more relevant knowledge on phenolic pro or antioxidant properties. Finally, it is not clear whether authors did 6 experiments on plasma that was pooled or indeed they did all experiments related to antiox properties with individual plasma specimens. This should be clarified.
- Methodology 3 – 3. Preparation and of the phenolic preparations from sea buckthorn twigs and leaves
Please optimize writing in regard of repetition. The same procedure was used for leaves and twigs. Therefore, after explaining the protocol for leaves, only address specific differences in the protocol of phenolic extracts from twigs.
- Results 1– Figures 4 and 5 have too many shades of grey and therefore difficult to grasp. Better use colour that could reflect plant parts from whom phenolics stem, like 3 green shades for leaves and 3 brownish shades for twigs. Suggestion to insert percentages of inhibition in the Table 1.
- Results 2 - instead of Figure 1 with chromatograms that could become supplementary figure, create a table that will describe content specifity of your novel phenolic preparations.
- Results 3 – consider inputting individual values within bars in figures 4 and 5. For example to get something like in attachement.
- Discussion / it needs new structure, more meat, more focused discussion, more references to back up new ideas. It is not suitable to write:” Nevertheless, the changes observed here are not so easy to explain, since for instance, preparation A was rich in ellagotanins (Fig.1), while hydrolysable tanins and ellagic acid were also dominant compounds (above 31%) of previously tested LF [7].” I agree that sometimes we scientist need to be virtuous, when examining comprehensive areas, and I can understand authors, since I am and I was in similar situation. However, we have to be focused and brave and to be determined to give useful and concrete knowledge. I have seen excellent parts in the discussion, with high and correct associative manner such as:” On the other hand, we demonstrated that the lowest concentration of tested rhamnoides preparations inhibiting microbial growth (62 μg/mL - MIC A against S. epidermidis) was still higher than the concentrations in which these preparations were able to reduce lipid peroxidation induced by H2O2/Fe and protect plasma proteins against H2O2/Fe – induced protein carbonylation (10 and 50 μg/mL, Fig. 3-4). Thus, tested preparations could be safely administered orally as antioxidants without affecting gut microbiota.” Therefore, I believe that authors are capable to rewrite discussion to give us clearer, useful info on the topic that they have study, to justify why they did it and what is that novel knowledge that they brought to us.
Minor Comments:
- No row numbering, which makes review process more difficult.
- Title – try to create more informative, relevant and thus more impactful.
- Insert reference after the first sentence in paragraph 2.3
- Instead of extracted with hexane, it is better to use defatted with hexane
- What was the v/v ratio of liquid preparation of phenolics: plasma in 2.8 paragraph?
- “3.1. Chemical characterization of the tested s” something is missing…etc
Kind regards and lot of luck in your professional efforts.

Author Response
General Comment
The manuscript of Anna Stochmal, Bartosz Skalski, Rostyslav Pietukhov, Beata Sadowska, Joanna Rywaniak, Urszula Wójcik-Bojek, Lukasz Grabarczyk, Jerzy Żuchowski, Beata Olas *entitled: “A comparison of the biological properties of phenolic preparations with different chemical content from Elaeagnus rhamnoides (L.) A. Nelson leaves and twigs“ submitted to Molecules journal, presents their work in the field of natural products chemistry. Precisely, human blood plasma was used as in vitro model to study potential protective effects of differently produced polyphenolic preparations from leafs and twigs against lipid peroxidation and protein carbonylation events caused by excessive ROS (OH radical) production with H2O2/Fe2+ Fenton-like system. In addition, authors assessed survival of human foreskin fibroblasts in cell culture via MTT assay and microbiological resistance on several ATCC strains of pathogenic and normal skin flora bacteria and fungi. The sole novelty is related to the production of 3 different polyphenolic preparations from leaves and twigs each (6 extracts in total). The other methods and approaches were elaborated in their previous research, and were used here, which is important knowledge addition, if the aim was to decipher which classes of phenolic compounds are more responsible for certain bio-actions or effects under consideration.
However, there are some necessary improvements, related to brevity and clarity of manuscript, methodological issues and flow & quality of discussion.
Major Comments
- While the overall research design is appropriate, there is a room for improvement in terms of clarity, scientific methodology, data/results presentation and contribution to discussion via focused and brave synthetic approach. The best way to address this comment is to respond to all specific and minor comments outlined below.
Response: It is done.
- Style of the English language. The same effort that authors exerted by doing the experimental part, should be repeated when writing the manuscript. Frequently I face decent experimental works and setups, and poor writing. Inadequate writing negatively influences understanding (sense that was aimed to be conveyed to reading public), and meaning as well. For example, when expressing an idea in a sentence and then backing up with references, this expression should be specific as possible, informative and still brief. If possible, engage English language editor with life science background to correct grammar and style based on these recommendations and major study aim. Throughout the manuscript there are needed repairs, I give one example here, as the second sentence from discussion: “Considering direct antimicrobial activity of tested preparations from sea buckthorn leaves and twigs in the context on our previous study on biological activity of phenolic and non-polar fractions obtained from the same plant material [7] no spectacular differences were found.” It seems like you disrespect your own work. Instead it can be expressed like this:” Antimicrobial activity of our novel polyphenolic preparations made in sequential manner from the full extract was similar to antimicrobial effects exerted by polyphenolic leaf and twig extracts in our previous study (alternatively here could be mentioned the difference in preparation from the current study and the previous one, so that authors and reading public has a chance to conclude something sensibly).
Response: We have corrected.
Abstract should not be written as laboratory diary. A reader does not care about polyphenolic preparations labelled as A. B C d E F, but care to know how they are different and from what plant source they stem.
Response: A detailed description can be found later.
Introduction “Although the major components (especially phenolic compounds) of various organs of sea buckthorn have been described, biological properties of many of these phytochem-icals still remains unexplored. In this study, we focused on chemical composition and bi-ological activity of three phenolic preparations (A, B, and C), which were isolated from the extract of sea buckthorn leaves, and three phenolic preparations (D, E, and F), which were obtained from sea buckthorn twig extract. The objective was to investigate cytotox-icity of these preparations against human fibroblasts line HFF-1 using MTT reduction as-say, their anti- or prooxidant activity against the effect of a biological oxidant - H2O2/Fe (the donor of hydroxyl radicals) on human plasma lipids and proteins (measured by the level of thiobarbituric acid reactive substances (TBARS, the marker of lipid peroxidation and carbonyl groups – the marker of protein oxidation) in vitro. Antimicrobial activity of tested preparations against Gram-positive (Staphylococcus aureus, epidermidis, Enterococ-cus faecalis) and Gram-negative bacteria (Escherichia coli, Pseudomonas aeruginosa), as well as against fungi (Candida albicans, C. glabrata) by EUCAST approved broth microdilution method followed by growth on solid media was also assessed.”
It is not clear what was the reason that drove authors to conduct new study on the same parts and substrate, e.g. plant sea buckthorn leaves and twigs, with the same methodology. What was the rationale for the objective stated? What was the hypothesis? This should be rewritten. I already gave an idea how to do it, please see end of comment no 2. The rationale/hypothesis could be:
If very well characterized sub-fractions of phenolic extracts are made with contents conferring high specifities/uniqueness, than by assessing simultaneously their bio active properties such as pro or antioxidant and antimicrobial effects, it is possible to decipher the most impactful classes of phenolic compounds responsible for the effects observed.
Response: We have added hypothesis.
- Methodology 1– The most important concern here is weather authors used simultaneous incubation of human blood plasma with artificial Fenton-like system with H2O2/Fe2+ and polyphenolic extracts. If this is the situation, then they should read this publication “Enhanced Degradation of Phenol by a Fenton-LikeSystem (Fe/EDTA/H2O2) at Circumneutral pH” from MDPI Catalysts journal https://www.mdpi.com/2073-4344/9/5/474 and amend manuscript accordingly either with proper experimental control or by elaborating it in discussion based on their previous or others results. It might be that their results could be even more pronounced if there was no decomposition of certain polyphenolics caused by acidic to neutral pH in combination with 4.7 mM H2O2/3.8 mM Fe2SO4/2.5 mM EDTA.
Response: We have added more details about it: “Plasma was pre-incubated (5 min, at 37°C) with: preparations (A, B, and C) from the leaves of sea buckthorn at the final concentrations of 5-50 mg/mL and then 4.7 mM H2O2/3.8 mM Fe2SO4/2.5 mM EDTA (25 min, at 37°C); preparations (D, E, and F) from the twigs of sea buckthorn at the final concentrations of 5-50 mg/mL and then 4.7 mM H2O2/3.8 mM Fe2SO4/2.5 mM EDTA (25 min, at 37°C).
Moreover, this arrangement was used in our other publications:
- „Biological properties of Elaeagnus rhamnoides (L.) A. Nelson twig and leaf extracts”. Skalski, B. Kontek, B. Lis, B. Olas, Ł. Grabarczyk, A. Stochmal, J. Żuchowski. BMC Complementary and Alternative Medicine. 2019, 19(148), 1-12.
- „Phenolic fraction and nonpolar fraction from sea buckthorn leaves and twigs: chemical profile and biological activity”. Skalski, B. Kontek, B. Olas, J. Żuchowski, A. Stochmal. Future Medicinal Chemistry. 2018, 10(20), 2381-2394.
- „Isorhamnetin and its new derivatives isolated from sea buckthorn berries prevent H2O2/Fe – induced oxidative stress and changes in hemostasis”
Skalski, B. Lis, Ł. Pecio, B. Kontek, B. Olas, J. Żuchowski, A. Stochmal. Food and Chemical Toxicology. 2019, 125, 614-620.
- Methodology 2 – if authors stated that they aim to examine prooxidant effects of phenolics from leaves and twigs, why they haven’t included treatment comprised of solely plasma and phenolic extract incubated for 30 min on 37 degree and run TBARS test to look for certain prooxidant effects. There are of course other experimental options to do so. Please check paper https://www.hindawi.com/journals/jt/2011/467305/ for this venue. To be clear enough, the proper group setup beside those shown in figure 4 and 5 should be 1) only plasma, 2) plasma and phenolic extracts, 3) plasma with Fenton like system, 4) plasma with Fenton like system and extracts. Perhaps authors checked this aspect previously; if so certain amendments and elaborations should be included in the new manuscript version. Finally, authors could substantially improve their study by assessing individual human plasma with the most effective phenolic preparations, to contribute with more relevant knowledge on phenolic pro or antioxidant properties. Finally, it is not clear whether authors did 6 experiments on plasma that was pooled or indeed they did all experiments related to antiox properties with individual plasma specimens. This should be clarified.
Response: We have added more information about it: “Control negative (-) refers to plasma not treated with H2O2/Fe, whereas control positive (+) refers to plasma treated with H2O2/Fe.” (legend to figures 4 and 5). “Neither the preparations altered the level of biomarkers of oxidative stress in plasma not treated with H2O2/Fe (data not shown)” (chapter of results).
- Methodology 3 – 3. Preparation and of the phenolic preparations from sea buckthorn twigs and leaves
Please optimize writing in regard of repetition. The same procedure was used for leaves and twigs. Therefore, after explaining the protocol for leaves, only address specific differences in the protocol of phenolic extracts from twigs.
Response: We have corrected.
- Results 1– Figures 4 and 5 have too many shades of grey and therefore difficult to grasp. Better use colour that could reflect plant parts from whom phenolics stem, like 3 green shades for leaves and 3 brownish shades for twigs. Suggestion to insert percentages of inhibition in the Table 1.
Response: We have corrected Figures 4 and 5. However, we did not add percentages of inhibition in the Table 1. Because, Table 1 may only demonstrates comparative effects of the six preparations (A-F) from sea buckthorn leaves and twigs (at the used concentration – 10 µg/mL) on two selected parameters of oxidative stress: lipid peroxidation and protein carbonylation, in plasma treated with H2O2/Fe.
Results 2 - instead of Figure 1 with chromatograms that could become supplementary figure, create a table that will describe content specifity of your novel phenolic preparations.
Response: Such a table may be a quite good solution for presentation of the composition of the investigated sea buckthorn preparations. However, the preparations significantly differ in their composition, which is very easily visible on chromatograms. For this reason we propose to present them in the main manuscript, not in the supplementary data. In addition, chromatograms provide more data, as the relative content of individual compounds can also be evaluated.
- Results 3 – consider inputting individual values within bars in figures 4 and 5. For example to get something like in attachement.
Response: We have not added individual values (Fig. 4 and 5), because these figures will be difficult read.
- Discussion / it needs new structure, more meat, more focused discussion, more references to back up new ideas. It is not suitable to write:” Nevertheless, the changes observed here are not so easy to explain, since for instance, preparation A was rich in ellagotanins (Fig.1), while hydrolysable tanins and ellagic acid were also dominant compounds (above 31%) of previously tested LF [7].” I agree that sometimes we scientist need to be virtuous, when examining comprehensive areas, and I can understand authors, since I am and I was in similar situation. However, we have to be focused and brave and to be determined to give useful and concrete knowledge. I have seen excellent parts in the discussion, with high and correct associative manner such as:” On the other hand, we demonstrated that the lowest concentration of tested rhamnoides preparations inhibiting microbial growth (62 μg/mL - MIC A against S. epidermidis) was still higher than the concentrations in which these preparations were able to reduce lipid peroxidation induced by H2O2/Fe and protect plasma proteins against H2O2/Fe – induced protein carbonylation (10 and 50 μg/mL, Fig. 3-4). Thus, tested preparations could be safely administered orally as antioxidants without affecting gut microbiota.” Therefore, I believe that authors are capable to rewrite discussion to give us clearer, useful info on the topic that they have study, to justify why they did it and what is that novel knowledge that they brought to us.
Response: It is done.
Minor Comments:
- No row numbering, which makes review process more difficult.
Response: We have added.
- Title – try to create more informative, relevant and thus more impactful.
Response: It is done.
- Insert reference after the first sentence in paragraph 2.3
Response: We have added.
- Instead of extracted with hexane, it is better to use defatted with hexane
Response: We have corrected.
- What was the v/v ratio of liquid preparation of phenolics: plasma in 2.8 paragraph?
Response: TBARS (5:450), -CO (0,5:45), -SH (1:90).
- “3.1. Chemical characterization of the tested s” something is missing…etc
Response: We have corrected.
Reviewer 2 Report
Dear all,
below are my comments and suggestions:
The abstract should include methods and at least some significant results
Please indicate the bibliographic sources for Section 2.3. Preparation and of the phenolic preparations from sea buckthorn twigs and leaves
Please indicate the bibliographic sources for Section 2.7. Cytotoxicity of sea buckthorn preparations
Please indicate the bibliographic sources for Section 2.9.1. Plasma lipid peroxidation and Section 2.10. Antimicrobial activity
In the manuscript, it was used to express MIC / MBC values the units such as mg/ml, but also μg / ml units, please check.
The number of bibliographic sources is not adequate, less than 50% of the total bibliographic sources are from the last 5 years and a small number of bibliographic sources were used, please modify/add in all over the text. Also, more self-citations were used.
Please check your English (errors in spelling, grammar, and style) with a native English speaker.
Author Response
Dear all,
below are my comments and suggestions:
The abstract should include methods and at least some significant results
Response: It is done.
Please indicate the bibliographic sources for Section 2.3. Preparation and of the phenolic preparations from sea buckthorn twigs and leaves
Response: It is done.
Please indicate the bibliographic sources for Section 2.7. Cytotoxicity of sea buckthorn preparations
Response: Self modified MTT method was used based on methodology provided by Müller and Kramer, Journal of Antimicrobial Chemotherapy (2008) 61, 1281–1287, doi:10.1093/jac/dkn125 and EN ISO 10993-5 for cytotoxicity testing. Appropriate references have been included in revised text.
Please indicate the bibliographic sources for
Section 2.9.1. Plasma lipid peroxidation
and Section 2.10. Antimicrobial activity
Response: There is bibliographic sources in section 2.9.1. Appropriate bibliographic source (EUCAST) for 2.10. methodology has been added to the revised text.
In the manuscript, it was used to express MIC / MBC values the units such as mg/ml, but also μg / ml units, please check.
Response: The units “mg/mL” in Table 2 were introduced to avoid “zeros” and the results be more clear (visible). In description of the results in the text we used “μg/mL” for simple compare with other results (e.g. cytotoxicity, antioxidative activity). However, according to Reviewer suggestion the units have been unify to μg/mL in revised text.
The number of bibliographic sources is not adequate, less than 50% of the total bibliographic sources are from the last 5 years and a small number of bibliographic sources were used, please modify/add in all over the text. Also, more self-citations were used.
Response: We have added.
Please check your English (errors in spelling, grammar, and style) with a native English speaker.
Response: It is done.
Reviewer 3 Report
In general is a paper which is not very well written and need to be more focused and undergo substantial corrections in order to be suitable for publication. The use of many method is not always contribute in correct and well-structured scientific work especially when the authors use methods like organic solvent extraction which are out of the updated approach to use green technologies for production of human oriented prodcuts.
Therefore I recommend major review and re-evaluation and you find heredown my remarks
Page 1. “ The antioxidant effect of sea buckthorn is due to the pres-ence of numerous antioxidants (vitamin C, phenolic compounds, including phenolic acids and flavonoids) present in various organs of this plant [1]”.
Please you must include the polysaccharides in the substances that produce antioxidant effects.
Page 1. “different fractions” it is better to put “different parts”
Page 2 &3 The description is very complicated and it is very difficult to be understood. So the extraction steps is better to be described in one or more flowcharts stepwise in order to be better understood. Also you must make this part of the paper more clear, as now it is not
Also there is a remark on the methods of extraction used in your work. Many hazardous materials as methanol, hexane, formic acid etc are used despite the fact nowadays there is a trend for “Green” extraction/prodution methods especially when the product is to be used for preparation of cosmetics. This , according to my opinion reduces the applicability and the value of your work.
Page 7. Why did you use human fibroblasts line HFF-1? Is there a specific reason please explain
Page 9. Table1. Why the preparation A has not any effect to lipid and protein peroxidation while it contains natural antioxidants? this has to be explained
Page 9 &10 Table 2 You MIC values >1 but this is not very informative as >1 means 10 but also it can be 100. You must give exact numbers.
Page 10 & 11. The discussion is erratic and not focused. This must be more focused. There is a confusion for the targeted application of the extracts. Are they supposed to be used for human consumption or for cosmetics?
Author Response
In general is a paper which is not very well written and need to be more focused and undergo substantial corrections in order to be suitable for publication. The use of many method is not always contribute in correct and well-structured scientific work especially when the authors use methods like organic solvent extraction which are out of the updated approach to use green technologies for production of human oriented prodcuts.
Therefore I recommend major review and re-evaluation and you find heredown my remarks
Page 1. “ The antioxidant effect of sea buckthorn is due to the pres-ence of numerous antioxidants (vitamin C, phenolic compounds, including phenolic acids and flavonoids) present in various organs of this plant [1]”.
Please you must include the polysaccharides in the substances that produce antioxidant effects.
Response: It is done.
Page 1. “different fractions” it is better to put “different parts”
Response: We have not corrected “different fractions”, because we used different fractions isolated from various parts of plant.
Page 2 &3 The description is very complicated and it is very difficult to be understood. So the extraction steps is better to be described in one or more flowcharts stepwise in order to be better understood. Also you must make this part of the paper more clear, as now it is not
Also there is a remark on the methods of extraction used in your work. Many hazardous materials as methanol, hexane, formic acid etc are used despite the fact nowadays there is a trend for “Green” extraction/prodution methods especially when the product is to be used for preparation of cosmetics. This , according to my opinion reduces the applicability and the value of your work.
Response: It has to be admitted that the described procedure of preparation of the investigated fractions is quite complicated. In the revised version, some unnecessary repetitions has been removed (e.g. in the case of the applied methods of preparation of the crude extracts from leaves and twigs, which were almost the same); moreover, a diagram of the extraction and fractionation procedures has been added, for convenience of readers.
As regards the green methods – all the experiments were performed within the frame a research project which involved isolation of secondary metabolites, including phenolic compounds from different parts of sea buckthorn. To make this goal easier, different organic solvents, such as hexane, formic acid, butanol etc. were applied. For example, hexane was used to remove unnecessary lipids and chlorophyll. We used methanol, instead of ethanol, because it was readily available in large amounts in our laboratory. However, our extraction and fractionation procedure may be easily adopted to the use of ethanol, to fulfill requirements of the (more) green production.
In any case, we could not use any kind of supercritical CO2 extraction, because of lack of a necessary equipment. Moreover, this method seems to be more suitable for less polar compounds, while many of sea buckthorn leaf and twig phenolics are quite hydrophilic ones.
Page 7. Why did you use human fibroblasts line HFF-1? Is there a specific reason please explain
Response: The fibroblasts are recommended as one of the lines for in vitro cytotoxicity testing in International Standard ISO 10993-5:2009(E) [reference 12]. Usually mouse fibroblasts line L929 are used, however because we assume the use of sea buckthorn preparations as potential biologically active substances in pharmacologic or cosmetic products, we decided to use human fibroblasts (line HFF-1) for cytotoxicity testing. This short explanation has been included in the revised text (section 3.2.).
Page 9. Table1. Why the preparation A has not any effect to lipid and protein peroxidation while it contains natural antioxidants? this has to be explained
Response: We have added more information about it in the chapters of results and discussions.
Page 9 &10 Table 2 You MIC values >1 but this is not very informative as >1 means 10 but also it can be 100. You must give exact numbers.
Response: As it was described in the text (Results), MIC/MBC >1 mg/ml (>1000 µg/ml) means that the preparations were inactive against bacteria/fungi in whole range of concentrations tested, so the results are informative. Of course, then MIC/MBC may be 10 or 100 mg/mL, but it does not matter, since such high concentrations you will never obtain in the host tissues because of bioavailability and cytotoxicity. So, if you assume the potential use of tested preparations for pharmacologic or cosmetic purpose the effective concentrations should not exceed 1 mg/ml (1000 µg/ml) for complex preparations (e.g. extracts) and 0.5 mg/ml (500 µg/ml) for pure compounds.
Page 10 & 11. The discussion is erratic and not focused. This must be more focused. There is a confusion for the targeted application of the extracts. Are they supposed to be used for human consumption or for cosmetics?
Response: The tested extracts could be used both: as a dietary supplement and as an addition to cosmetics.
Round 2
Reviewer 2 Report
Dear all,
I recommend publishing the manuscript in this form.
Author Response
Thank you for your opinion.
Reviewer 3 Report
The following points are not correctly answered.
Page 9. Table1. Why the preparation A has not any effect to lipid and protein peroxidation while it contains natural antioxidants? this has to be explained
Response: We have added more information about it in the chapters of results and discussions. Where???
Page 9 &10 Table 2 You MIC values >1 but this is not very informative as >1 means 10 but also it can be 100. You must give exact numbers. why not
Response: As it was described in the text (Results), MIC/MBC >1 mg/ml (>1000 µg/ml) means that the preparations were inactive against bacteria/fungi in whole range of concentrations tested, so the results are informative. Of course, then MIC/MBC may be 10 or 100 mg/mL, but it does not matter, since such high concentrations you will never obtain in the host tissues because of bioavailability and cytotoxicity. So, if you assume the potential use of tested preparations for pharmacologic or cosmetic purpose the effective concentrations should not exceed 1 mg/ml (1000 µg/ml) for complex preparations (e.g. extracts) and 0.5 mg/ml (500 µg/ml) for pure compounds.
Page 10 & 11. The discussion is erratic and not focused. This must be more focused. There is a confusion for the targeted application of the extracts. Are they supposed to be used for human consumption or for cosmetics?
Response: The tested extracts could be used both: as a dietary supplement and as an addition to cosmetics.
The authors failed to provide the required replies so i am forced to recommend rejection
Author Response
The following points are not correctly answered.
Page 9. Table1. Why the preparation A has not any effect to lipid and protein peroxidation while it contains natural antioxidants? this has to be explained
Response: We have added more information about it in the chapters of results and discussions. For example, At the highest used concentration – 50 µg/mL, all used preparations from leaves and twigs, including the preparation A caused statistically significant reduction of lipid peroxidation induced by H2O2/Fe, as compared to the control sample (plasma treated only with H2O2/Fe) (Fig. 4). Moreover, the preparation A was shown to protect plasma proteins against H2O2/Fe – induced protein carbonylation, however, this effect was not statistically significant (Fig. 5). Table 1 only demonstrates comparative effects of the six preparations (A-F) from sea buckthorn leaves and twigs (at the used concentration – 10 µg/mL) on two selected parameters of oxidative stress. (the chapter of results).
Our present findings confirm that the six tested preparations (A-F, especially at the highest used concentration - 50 µg/mL) isolated from sea buckthorn leaves and twigs demonstrated antioxidative potential in an in vitro human plasma treated with H2O2/Fe: tested preparations (A-F) inhibited plasma lipid peroxidation induced by H2O2/Fe, as measured by the level of TBARS. Moreover, the tested preparations (A-F) reduced plasma protein carbonylation induced by H2O2/Fe. However, they had different influence on the oxidative stress in human plasma treated with H2O2/Fe, which may be attributed to the differences in their chemical content. Four preparations seem to offer the most promise (one preparation from leaves (preparation C) and three preparations from twigs (D-F)). (the chapter of discussion).
Page 9 &10 Table 2 You MIC values >1 but this is not very informative as >1 means 10 but also it can be 100. You must give exact numbers. why not
Response: As it was described in the text (Results), MIC/MBC >1 mg/ml (>1000 µg/ml) means that the preparations were inactive against bacteria/fungi in whole range of concentrations tested, so the results are informative. Of course, then MIC/MBC may be 10 or 100 mg/mL, but it does not matter, since such high concentrations you will never obtain in the host tissues because of bioavailability and cytotoxicity. So, if you assume the potential use of tested preparations for pharmacologic or cosmetic purpose the effective concentrations should not exceed 1 mg/ml (1000 µg/ml) for complex preparations (e.g. extracts) and 0.5 mg/ml (500 µg/ml) for pure compounds.
Moreover, to test the concentration of 100 mg/ml, the starting solution of the extract in DMSO would have to be at a concentration of 4000 mg/ml (4 g/ml !!!), provided that such a solution could be prepared at all (solubility!). Therefore, we have not done.
Page 10 & 11. The discussion is erratic and not focused. This must be more focused. There is a confusion for the targeted application of the extracts. Are they supposed to be used for human consumption or for cosmetics?
Response: In the chapter of discussion, we have described antimicrobial and antioxidant potential of tested preparations, and we also suggested that the tested extracts could be used both: as a dietary supplement and as an addition to cosmetics.